# Symmetry in category systems across languages

Charles Kemp ✉

Language reflects how people organize experience into categories, and cross-linguistic comparison can help to identify general principles that shape categorization. Here we argue that symmetry is one such principle, and present a symmetry-based theory that predicts whether category systems for a given domain tend to include an even or an odd number of categories. We test the theory against cross-linguistic data previously compiled for a range of domains and find that deictic day-naming and tense-marking systems tend to have an odd number of categories, but that systems for domains including seasons, phases of the moon, kinship, and cardinal directions tend to have an even number of categories. Our results therefore provide evidence of the widespread influence of symmetry on categorization across languages and domains.

Anyone familiar with two or more languages will know that terms for kin types, colors, seasons, animals, and many other domains often do not translate perfectly across languages. Despite this variation, scholars have identified universal principles that appear to shape categorization across cultures. Named categories tend to be useful for communication[1,2] and relatively easy to learn and remember[3,4], and properties that support these goals include perceptual salience[5], convexity[6], compositionality[7] and hierarchical organization[3]. We propose that symmetry is another such property, and show that the principle of symmetry supports predictions about whether category systems across domains tend to have an even or an odd number of categories.

Symmetry is evident in day-naming systems that treat the future (e.g., *tomorrow*) similarly to the past (*yesterday*), in season-naming systems that give balanced treatment to the hottest (*summer*) and coldest (*winter*) periods of the year, and in directional systems that label both ends of an axis (e.g., *east* and *west*). Many such symmetries reflect symmetries in the world, such as predictable seasonal cycles, but they also suggest a cognitive inclination towards symmetry. Symmetric category systems are compressible[8], which makes them relatively simple[2] and thus relatively easy to learn and remember. Previous researchers have proposed that category systems are shaped by evolutionary pressures toward simplicity[2,8,9], and our proposed symmetry preference can be seen as a special case of this more general proposal.

Our work is related to previous investigations of the role of symmetry in perception and cognition[10–13], and in particular to previous approaches that characterize the simplicity or goodness of a category in terms of structural properties such as symmetry[14]. Most work in this tradition focuses on visual objects such as patterns of dots, but ideas about symmetry and invariance have also been used to characterize the goodness of categories defined in terms of abstract binary features[15]. We go beyond previous psychological work on symmetry and categorization by considering the symmetry of entire systems of categories, and by using symmetry to make predictions about categorization across languages.

Our work also builds on and extends the structuralist approach in anthropology, which has been applied to domains including food, mythology, and kinship. Levi-Strauss[16](p 2) describes structuralism as the "quest for the invariant, or for the invariant elements among superficial differences," and invariance is closely related to symmetry. A standard idea in group theory, the mathematical theory of symmetry, is that symmetry can be defined as invariance with respect to a set of transformations, and our theory makes use of this definition. Our approach is therefore highly compatible with structuralism, but we go beyond prior structuralist approaches by evaluating our theory against cross-linguistic data from a diverse set of domains. The structuralist tradition has been criticized for placing more emphasis on theory than data[17], but our work highlights the fact that structuralist approaches can make testable empirical predictions.

Each domain we consider is represented by a structure, and four examples are shown in Fig. 1. A structure includes a space along with a set of transformations defined over that space. We introduce our

Melbourne School of Psychological Sciences, University of Melbourne, Victoria, Australia. ✉e-mail: c.kemp@unimelb.edu.au

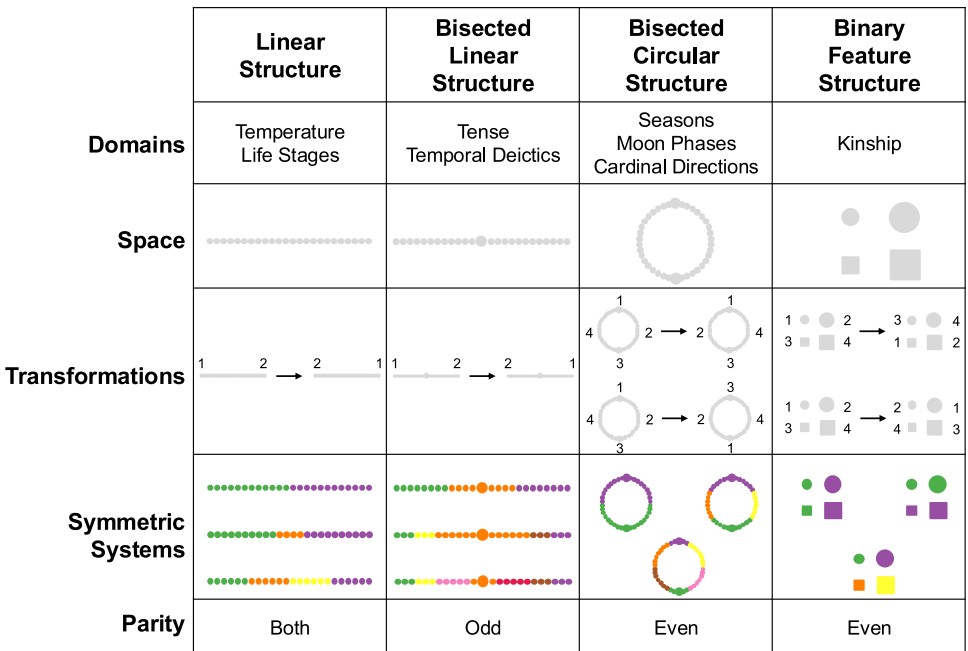

**Fig. 1 | Illustration of how symmetry applies to four different structures.** Each structure consists of a space along with a set of transformations of that space. The transformation for the linear and bisected linear structures inverts the one-dimensional space. The transformations for the bisected circular structure are a reflection in the vertical axis and a rotation through 180°, and the transformations for the binary feature structure exchange the values of the two features. A category system is symmetric if invariant under all transformations shown, and three symmetric systems are shown for each structure. The top row shows examples of domains that can be modeled using each structure, and the bottom row indicates whether semantic systems in those domains are expected to have both parities or to show a preference towards odd or even sizes.

theory using the linear structure as an example, and the remaining three structures are introduced in subsequent sections.

The linear structure includes a one-dimensional space and is appropriate for systems of temperature terms (e.g., English *cold*, *cool*, *warm* and *hot*), where the underlying dimension is temperature[18]. Each structure may be relevant to multiple semantic domains, and the linear structure is also appropriate for systems of life-stage terms (e.g., English *infant*, *toddler*, *child*, *teenager*, *adult*), where the underlying dimension is age[19].

A category system organizes the points in the space into categories, and Fig. 1 shows examples that partition the one-dimensional space into two, three, and four categories. All of these categories are connected, and we will assume that any category within a connected space must also be connected. For example, we do not allow a category that includes both ends of the one-dimensional space but excludes the central region.

All of the example systems in Fig. 1 are also *symmetric*, where symmetry is defined in terms of the transformations shown. For the linear structure in Fig. 1, there is a single transformation that inverts the one-dimensional space so that the left endpoint maps to the right endpoint. A category system is symmetric if it is invariant under all transformations associated with a structure. For example, inverting the first example system shown for the linear structure affects the category labels ('purple' is now on the left rather than the right), but the resulting system is isomorphic to the original. A more formal definition of symmetry is provided in the Methods.

We will say that a structure is *odd* if all symmetric category systems defined over that structure have an odd number of categories. Similarly, the structure is *even* if all symmetric category systems (except the system that assigns all points to the same category) have an even number of categories. If the symmetric systems for a structure include both odd and even examples (as is the case for the linear structure in Fig. 1), we will say that the structure has both parities.

Our core theoretical proposal is that human languages show a preference for symmetric category systems. If so, then systems for domains with an odd structure should tend to have an odd number of categories, and attested systems for even domains should tend to have an even number of categories. If a structure has both parities (such as the linear structure in Fig. 1), then our theory suggests that both odd and even systems are expected, and by default should have relatively similar frequencies. We test these predictions here using a diverse collection of cross-linguistic data sets that were mostly compiled by previous scholars and are archived at Zenodo[20]. We initially focus on domains for which our theory predicts a tendency towards either even or odd systems, and test the prediction that domains with a bisected linear structure show a preference for odd-numbered systems, and that domains with bisected circular or binary feature structures show a preference for even-numbered systems. A subsequent section considers two control domains that are expected to show no preference towards either even or odd systems.

## Results

A bisected linear structure includes a one-dimensional space bisected by a privileged central point that must be assigned to a category. Figure 1 shows that the single transformation associated with the structure inverts the dimension about its central point. A bisected structure is similar to a linear structure, and the key difference between the two is whether the one-dimensional space has a privileged central point. We consider two temporal domains that are instances of the bisected linear structure. In both cases, the dimension is time, and the privileged point represents the present, or the point that separates past from future. The first domain is deictic day-naming: for example, English includes the terms "yesterday," "today," and "tomorrow," and many languages also have terms for the day before yesterday and the day after tomorrow. The second domain is tense marking: for example, English distinguishes between past, present and future using

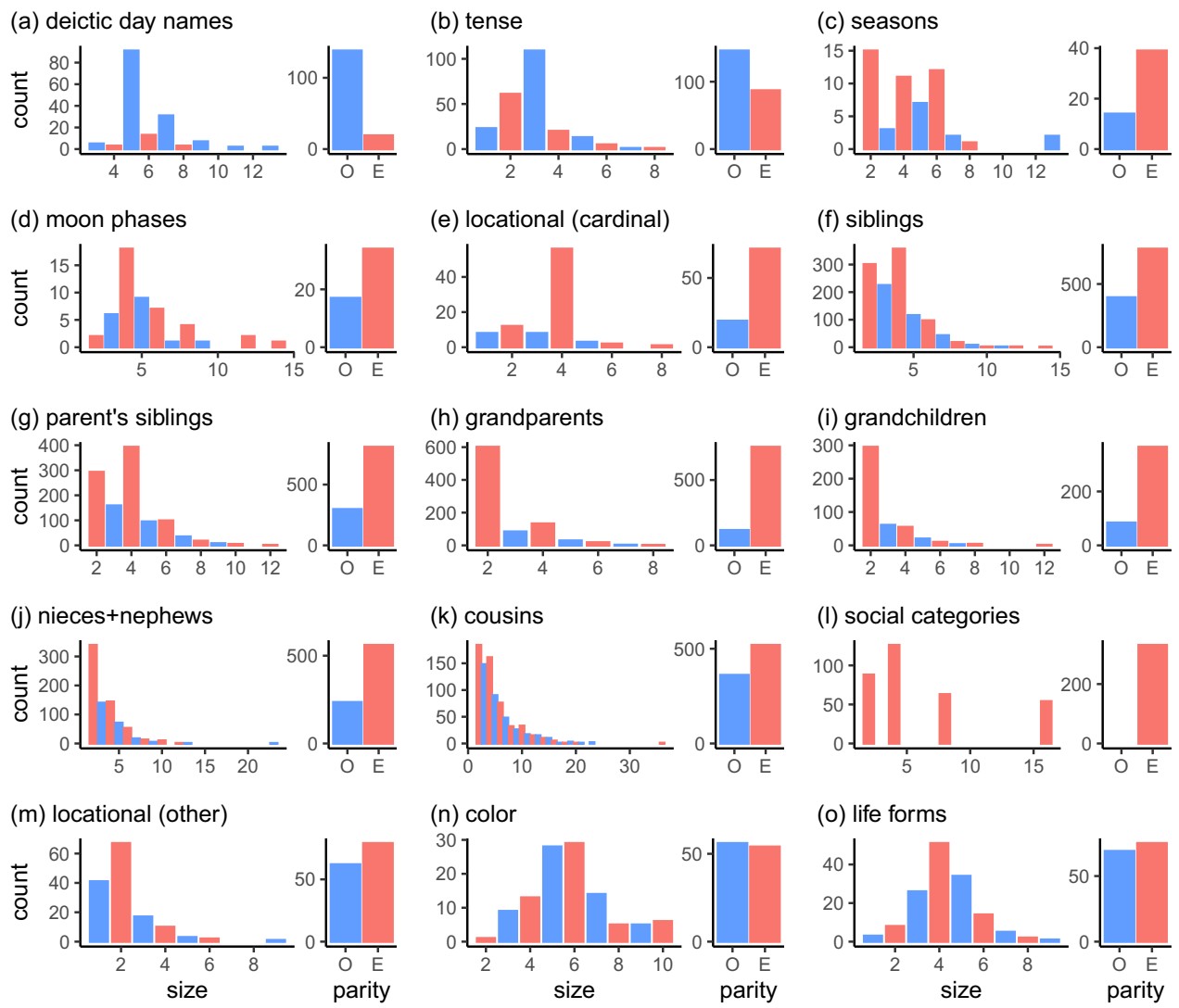

**Fig. 2 | Analysis of system sizes across 15 domains and subdomains.** Each panel shows a distribution of system sizes across languages (left), and a summary plot showing the number of odd (O, blue) and even (E, red) systems (right). Panels (**a**) and (**b**) show a preference for odd systems, panels (**c**) through (**m**) show a preference for even systems, and panels (**n**) and (**o**) show a roughly equal division across odd and even systems.

inflectional morphology (e.g., "she walked", "she walks") and periphrasis ("she will walk").

In both of these temporal domains, symmetric systems must have an odd number of categories. Intuitively, a symmetric system will have one category that includes the present, and each category for the past will be mirrored by a corresponding category for the future, which leads to an odd number of categories overall.

### Deictic day names

We analyzed a set of 157 day-naming systems compiled by Tent[21]. Distributions of these systems across areas and language families are reported in Tables S1, S2 of the Supplementary Methods. The distribution over the sizes of these systems is shown in Fig. 2a, and counts for odd and even systems are shown in the same panel. The most common system has a size of 5 and picks out two days on either side of 'today', and systems of size 7 are also relatively common. The sawtooth shape of the size distribution (high for odd sizes and lower for intervening even sizes) reveals a strong preference for odd-numbered systems across languages.

Tent's day-naming data are not balanced across language families, and the size distribution in Fig. 2a does not take genetic relatedness

into account. To estimate the prevalence of odd-numbered systems while controlling for genetic relatedness, we implemented an intercept-only phylogenetic regression model[22] with system parity as the dependent variable. For this and other regression models reported in the paper, 95% credible intervals on the intercepts are reported in Fig. 3, and analogous results showing frequentist confidence intervals are reported in Fig. S5. We take a credible interval that excludes and lies to the left of 0.5 as strong evidence that a domain favors odd systems, and an interval that excludes and lies to the right of 0.5 as strong evidence that a domain favors even systems. In Fig. 3, the credible interval for day-naming lies to the left of 0.5, indicating that this domain shows a strong preference for odd systems.

### Tense

Next, we analyzed a database of 233 tense systems compiled by Velupillai[23]. The size distribution in Fig. 2b shows that the most common size is three, and the credible interval in Fig. 3 excludes 0.5, suggesting a reliable preference for odd-numbered systems. The size distribution in Fig. 2b, however, is relatively smooth and does not have the sawtooth shape that provides the strongest evidence for a parity preference. Further reason for caution is provided by systems with

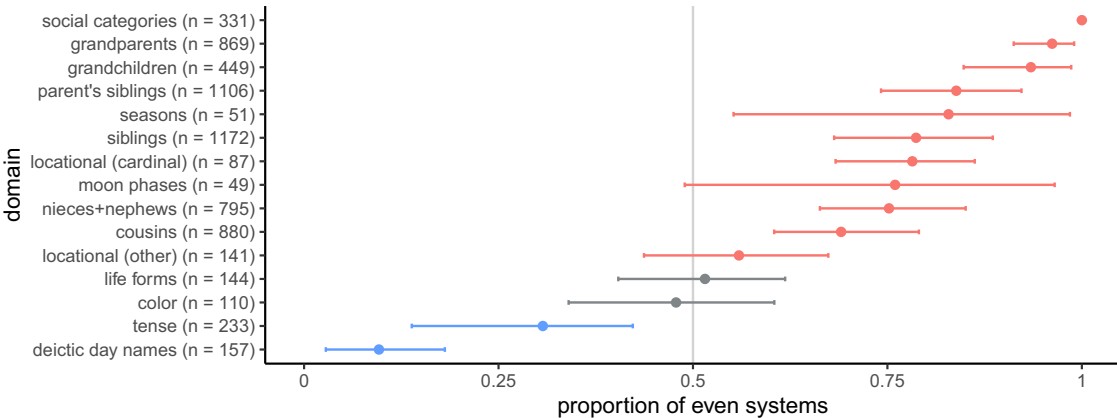

**Fig. 3 | Estimates of the probability that a system is even, along with 95% credible intervals on the intercept of a logistic regression model (locational domains) or a phylogenetic regression model (all other domains except social categories).** Point estimates are posterior means, and a logistic transform has been applied so that an intercept of 0 corresponds to a probability of 0.5. Credible intervals for domains predicted to be even or odd are shown in red or blue, respectively, and credible intervals for control domains are shown in gray. High estimates indicate a strong preference for even rather than odd-numbered systems. The values of *n* along the y-axis show the number of category systems used for each domain.

remoteness distinctions that divide either the past or future or both into two or more categories. Remoteness distinctions are more likely to subdivide the past than the future[23,24], and Velupillai[23] notes that symmetric remoteness distinctions are relatively rare. Overall, then, tense-marking does show a preference for odd-numbered systems, but the role of symmetry is less apparent for tense marking than for deictic day-naming.

## Seasons

Consider now the bisected circular structure shown in Fig. 1. The space for this structure is a circle that is bisected by a privileged axis, leading to two privileged points. Unlike bisected linear structures, bisected circular structures are predicted to favor systems with an even number of categories. We consider three domains with bisected circular structures: season naming, names for phases of the moon, and names for cardinal directions.

Consider a simple seasonal cycle in which a single variable (temperature, say) varies from low to high and back to low over the course of a year. The bisected circular structure in Fig. 1 can capture this seasonal progression, and the two privileged points could be the summer and winter solstices, or the two equinoxes (see Supplementary Discussion for additional discussion). The two transformations in Fig. 1 correspond to reversing the temporal order of the cycle and shifting the cycle by half a year.

The Western season system divides the circle into four arcs corresponding to winter, spring, summer, and autumn, but other systems of season categories are attested across cultures. We analyzed 53 season systems compiled by Kemp et al.[25], and the size distribution for these systems (Fig. 2c) shows a sawtooth pattern that favors even systems. The credible interval for seasons in Fig. 3 excludes 0.5, indicating a robust preference for even systems. A second database of season systems was compiled by Orlove[26], and 23 out of the 28 systems in this database are even. Orlove notes that odd-numbered systems are rare, but did not provide an explanation of this result. Our approach suggests that this result is a special case of a more general preference for symmetric category systems.

## Moon phases

Consider now a lunar cycle that begins with a new moon, which waxes to become a full moon, then wanes to return to the start of the cycle. The two privileged points in this cycle may correspond to the new moon and the full moon, and the transformations in Fig. 1 correspond to reversing the temporal order of the cycle and shifting the cycle by half a lunar month.

English has labels for four major phases of the lunar cycle (*new moon*, *first quarter*, *full moon* and *third quarter*, but the number of labeled phases varies across cultures. We collected a set of 52 moon-phase naming systems from the Human Relations Area Files[27] and a range of other sources. The distribution of system sizes is shown in Fig. 2d and reveals a preference for even systems driven primarily by a preference for systems of size four. The credible interval for moon phases, however, includes 0.5, indicating that the predicted preference for even systems does not have strong support. Some systems are odd because they use the same term for the first and third quarters, and others because they include two terms for the start of the lunar cycle (one for the new moon and another for the darkness preceding the new moon), but only one term for the midpoint of the cycle (full moon).

The lunar cycle leads to a monthly tidal cycle that includes *spring tides* near the new and full moons and *neap tides* near the first and third quarters. Many languages name parts of this monthly cycle, and our approach predicts that these monthly tide-naming systems will tend to include an even number of terms. Languages also have names for parts of the diurnal tidal cycle that moves between high and low tides every lunar day, and again, our theory predicts that these diurnal tide-naming systems will tend to include an even number of terms.

## Cardinal directions

Names for seasons, moon phases, and tides label processes that unfold through time, and the bisected circular structure in Fig. 1 seems appropriate for these and other temporal cycles. The structure, however, is not limited to temporal phenomena and can also be used to capture points of the compass. In this case, the privileged points may correspond to the directions of sunrise (east) and sunset (west), and the two transformations correspond to a reflection in the east-west axis and a rotation of 180°.

English labels four main cardinal directions (*north*, *south*, *east* and *west*), but this system is not universal. We analyzed a set of 90 cardinal directional systems from a database of Australian directional systems compiled by Hoffmann et al.[28]. The distribution over the sizes of these systems is shown in Fig. 2e, and reveals a strong preference for systems of size 4. There is also the hint of a sawtooth pattern, because there are more systems of sizes 2 and 4 than systems of size 3.

## Kinship

All of the structures considered so far incorporate continuous spaces, but our framework can also accommodate discrete feature spaces. Feature representations have been considered for domains including

personal pronouns[29], indefinite pronouns[30], spatial demonstratives[31], and spatial adpositions[32], and we consider several examples drawn from the domain of kinship.

The binary feature space in Fig. 1 represents a set of four sibling kin types defined in terms of two binary features: sex (male vs female) and relative age (older vs younger). The English kinship system ignores the relative age feature and organizes the kin types into two categories (*brother* and *sister*), but many other languages categorize these kin types differently. The two transformations in Fig. 1 exchange values of the two binary features: the first exchanges male and female, and the second exchanges older and younger. Any category system that is symmetric with respect to these transformations must have an even number of categories.

Figure 2f shows the size distribution over 1172 sibling systems derived from Kinbank[33]. Some of these systems include overlapping categories (e.g., *younger brother* and *brother*), and some are defined using features that go beyond the example in Fig. 1, such as a relative sex feature that indicates whether the referent of a sibling term has the same sex as the speaker[34]. As a result, a number of systems have sizes exceeding four. Figure 2f shows that even systems are preferred across languages, and that systems of size 3 are less common than both systems of size 2 and systems of size 4.

Figure 2 includes analogous results for five other subdomains: parents' siblings, grandparents, grandchildren, nieces and nephews, and cousins. The features needed to represent these subdomains vary: for example, the subdomain of grandparents requires a "sex of connecting relative" feature (paternal vs maternal) that is not needed for the subdomain of siblings. All of these subdomains, however, can be represented using a collection of binary features, which means that even systems are expected for all of these subdomains. Figure 3 indicates that all 6 kinship subdomains show a robust preference for even systems, and sawtooth patterns are evident for siblings, parents' siblings, grandparents, and cousins. The data are therefore consistent with the prediction that even systems are typical in the domain of kinship.

Figure 1 focuses on domains with especially simple structures, but we now demonstrate that our theory has implications that go beyond these four simple structures.

## Australian Aboriginal social classification

In communities with classificatory kinship systems, rules of marriage and descent are formulated in terms of some number of social categories (also called skin names, sections, or marriage classes)[35]. For example, in a four-category Kariera system, individuals in category A may only marry individuals in category B. Any children produced by such a marriage will belong to category C, and can only marry individuals belonging to category D. Capturing the structure of these systems was one of the early successes of the structuralist approach to anthropology[36], and Levi-Strauss[37] includes an appendix demonstrating that the structures of certain classificatory systems are isomorphic to mathematical groups.

Here, we consider one simple property of these systems. Suppose that the set of possible marriage relationships is invariant under a transformation that changes the sex of each individual from male to female or vice versa. If so, then the system must organize individuals into pairs of marriageable categories (e.g., members of categories A and B may marry each other), which means that the total number of categories must be even. Figure 2l shows the size distribution over 331 systems classified by AustKin[35,38,39] as having sections, sub-sections, or moieties. All of them are even, and all have sizes that are powers of two.

## Non-cardinal directional systems

We previously considered cardinal directional systems, but many languages have other kinds of directional systems, including systems based on local rivers (*upstream* vs *downstream*), local coasts (*coastal* vs *inland*), wind direction (*upwind* vs *downwind*), and topographic landmarks (*uphill* vs *downhill*). Although these locational systems do not have the bisected circular structure used in our analysis of cardinal directional systems, they can often be represented using pairs of opposing concepts, such as *upstream* and *downstream*. If these systems are symmetric with respect to transformations that exchange these pairs, then both concepts need to be named, which means that symmetric locational systems will have an even number of categories.

Figure 2m shows the size distribution over 141 non-cardinal systems drawn from the database of Australian directional systems mentioned earlier[28]. We included systems based on rivers, coasts, wind direction, path of the sun, and tides, but excluded topographic systems because it was difficult to establish the sizes of these systems. Some languages include multiple non-cardinal systems, and these systems are included as separate entries in our data set. Figure 2m shows that the most common system size is 2, but overall, there is not a strong preference for even-numbered systems. Several systems label only one member of an opposition, perhaps because the other member is taken as a default. For example, Kuku Yalanji has a form *ngubar* for 'other side of the river', but appears to have no form for 'same side of the river'[40].

## Control domains

We now turn to two domains for which a strong preference for even or odd systems is not expected. The first is color. Color can be roughly modeled using a three-dimensional solid in which the axes correspond to red-green, blue-yellow, and light-dark. Psychophysical work suggests that color space has no true symmetries, but transformations that invert the three axes (e.g., that map red to green) are approximate symmetries of color space[41]. Even if these approximate symmetries are allowed, the Supplementary Methods show that symmetric color systems can be either even or odd, which means that no strong parity preference is expected. Figure 2n shows the size distribution over 110 color-naming systems drawn from the World Color Survey[42]. The distribution is relatively smooth and is almost perfectly balanced between odd and even systems.

The second control domain is folk biology, and we consider systems of life-form terms such as *bird*, *fish* and *snake*. The space of living creatures has a branching structure induced by evolution, but appears to have no symmetries that should favor either odd-numbered or even-numbered category systems. Figure 2o shows the size distribution over 144 systems of life form categories compiled by Brown[43]. As for color, the distribution is relatively smooth and shows no preference towards either odd or even systems.

Additional control domains could be considered, including domains with the linear structure in Fig. 1 (e.g., temperature terms[18] and life-stage terms[19]), and domains with no apparent symmetries (e.g., smell terms[44]). In all of these cases, we expect a relatively even balance between odd-numbered and even-numbered systems.

## Discussion

Our theory successfully predicts whether category systems across a number of domains tend to be even or odd, but with one exception (Australian social classification systems), our results show these predictions to be statistical tendencies rather than absolute universals[45]. Many individual languages are inconsistent with these predictions, and investigating these cases is likely to reveal cultural and functional factors that can override a default symmetry preference. For example, tense systems are more likely to subdivide the past than the future, and this finding is consistent with the observation that speakers are more likely to talk about the past than the future[46].

We showed that symmetry makes testable predictions about the parity of category systems, and focusing initially on parity offers two key advantages. First, parity can be easily compared across domains, as

we have done here. Second, parity is often easier to determine than other aspects of a category system, such as category boundaries or the underlying features or dimensions that are used to mentally represent categories. For example, ethnographic accounts of season naming often provide relatively clear information about the number of season categories used by a community, but the boundaries between these categories can be hard to establish, and it is not always clear which features (e.g., time of year, climate, ecological features) are used to define seasons. Ultimately, though, investigations of the influence of symmetry on categorization will need to go beyond parity alone, and the Supplementary Methods illustrate how symmetry can be investigated in domains for which symmetry makes no prediction about parity.

Our theory assumes that each domain has an underlying structure that is shared across cultures, but more work is needed to probe the nature of this structure and to assess whether it is truly universal. For seasons, lunar phases and cardinal directions, the underlying structure is hypothesized to include privileged points, and the Supplementary Discussion provides some reasons to think that these privileged points may vary across cultures. Different assumptions about the underlying structure can be tested for compatibility with cross-linguistic data, and lab experiments could also be developed to explore whether some points in a cycle (e.g., the absence of the moon in the lunar cycle) are more psychologically salient than others. The structures proposed here should therefore be viewed as working hypotheses that can potentially be refined by future studies.

We argued that symmetry influences how domains are carved up into categories, but symmetry can also be found in the linguistic forms used to label those categories. Tent[21] provides an extensive discussion of the forms used for day-naming and notes that many systems of forms show morphological or lexical symmetry. For example, Wayan attaches numerals (e.g., *rua* 'two') to the base *bogi* 'night', so that the day before yesterday is *a bogirua* and the day after tomorrow is *ei bogirua*. Similarly, speakers of Sa'a (Solomon Islands) name seven lunar phases in the first half of the lunar month, and the same names are used in reverse order with a different prefix during the second half[47]. Examples like these illustrate how morphological and lexical symmetry can provide direct evidence of the influence of symmetry on categorization.

Day-naming systems also hint that linguistic forms can reflect more abstract kinds of symmetry. Consider, for example, the German day-naming system, which distinguishes *vorgestern* (day before yesterday), *gestern* (yesterday), *heute* (today), *morgen* (tomorrow) and *ubermorgen* (day after tomorrow). Even though this system does not reuse morphemes for reference to both past and future, two related kinds of symmetry are evident. On both sides of *heute* (today), forms for remote days (*vorgestern* and *ubermorgen*) are created by adding prefixes to forms for less remote days (*gestern* and *morgen*). As a result, day names further from the present are labeled with longer forms, and this naming pattern is iconically related to a timeline centered on the present. Future studies can explore the extent to which form-based symmetries are found in domains other than day-naming, and how these symmetries relate to factors such as iconicity and compositionality.

Our theory does not characterize the evolutionary process by which symmetric category systems emerge, but makes an intriguing prediction about the trajectories produced by this process. Existing evolutionary accounts often suggest that categories are added one by one[48,49], but if the influence of symmetry is strong, then categories may be added in pairs, and this phenomenon may be especially likely when compositional forms are used. For example, season-naming trajectories might leap from a two-category stage to a four-category stage without passing through an intermediate three-category stage, and day-naming trajectories might leap from a three-category stage to a five-category stage. This evolutionary prediction could be tested in the lab[8,9] and phylogenetic analyses may also allow it to be tested against cross-linguistic data[50]. The same prediction could also be explored developmentally: for example, both English and German children learn *tomorrow* before *yesterday*[51,52], but it seems possible that German children learn *vorgestern* (day before yesterday) and *ubermorgen* (day after tomorrow) at around the same time[53].

Although we found that category systems tend to be symmetric, our results do not reveal whether the ultimate source of this symmetry lies in the world or the mind. A world-centered view proposes that symmetries in categorization are a faithful reflection of symmetries in the world, and may be appropriate for domains such as seasons and lunar phases, where symmetric category systems may arise as a direct reflection of regular variation in environmental phenomena. A mind-centered view proposes that symmetries in categorization reflect a cognitive bias towards symmetry, and may be appropriate for domains such as social classification, where categories appear to be socially constructed. The two views are not mutually exclusive, and future laboratory studies could assess their relative merits by asking whether the category systems acquired by learners tend to be more symmetric than the systems provided as input[54].

## Methods

We begin by providing a formal definition of symmetry, then provide detailed information about the data sets analyzed for each domain.

### Parity and domain structure

Let $\mathcal{S}$ be a space and $\mathcal{T}$ be a set of transformations acting on that space. Space $\mathcal{S}$ may be connected (as for the first three examples in Fig. 1) or not (as for the binary feature space in Fig. 1). Each transformation $t$ in $\mathcal{T}$ is a bijection $t : \mathcal{S} \to \mathcal{S}$ from $\mathcal{S}$ to itself.

A category system $\mathcal{C}$ is a partition of the space, and can be represented as a binary relation $R_C : \mathcal{S} \times \mathcal{S} \to \{0, 1\}$ that indicates whether or not two items belong to the same category. If $\mathcal{S}$ is connected, then we assume that all categories in $\mathcal{C}$ are also connected. A category system is symmetric if it is left unchanged by applying each transformation in $\mathcal{T}$. More formally, a symmetric system $\mathcal{C}$ satisfies

$$R_C(x, y) = R_C(t(x), t(y)) \qquad (1)$$

for each transformation $t$ in $\mathcal{T}$.

The approach just described can be characterized in terms of a mathematical group acting on space $\mathcal{S}$. For the linear and bisected linear structures in Fig. 1, the transformation group is $D_2$, the dihedral group of order 2, which is equivalent to the cyclic group $Z_2$ of order 2. For the bisected circular and binary feature structures, the transformation group is $D_4$, the dihedral group of order 4.

Although we assumed that each category system $C$ is a partition of $\mathcal{S}$, the same approach applies to category systems with "holes," or items that are not assigned to any category. Our approach can also be generalized to systems with overlapping categories, including hierarchical category systems.

### Domains

For each domain, we aimed to count the number of named categories per language. Some categories have multiple names, and it is sometimes hard to tell whether two names should be treated as labels for the same category or labels for different categories. For example, the database of Hoffmann et al.[28] lists Jaru terms glossed as 'south,' 'to south,' 'south side' and 'south across water,' and we treated all of these terms as labels for the single category 'south'.

Some of our sources (especially older sources) are likely to be influenced by Western categories. For example, an author may elicit terms that correspond to the Western concepts of new moon, first quarter, full moon, and third quarter but fail to record additional terms for lunar phases. Our impression is that this issue arises more

a)

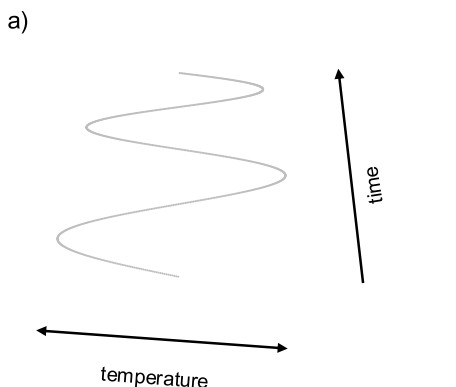

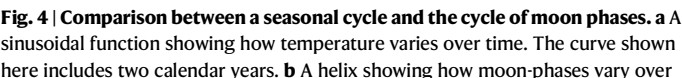

temperature

b)

time

**Fig. 4 | Comparison between a seasonal cycle and the cycle of moon phases. a** A sinusoidal function showing how temperature varies over time. The curve shown here includes two calendar years. **b** A helix showing how moon-phases vary over time for a viewer in the Northern Hemisphere. The curve shown here includes two and a half lunar months, and different colors indicate different months.

frequently for lunar phases and cardinal directional systems than for the other domains we consider, and may contribute to the finding that four is the most common size for both lunar-phase and cardinal systems.

Most of our regression analyses incorporated a phylogeny over language families, and all exceptions are noted below. Phylogenies for all domains were derived from Glottolog[55], and we used Glottolog data included in version 1.1.20 of the `lingtypology` package in R[56]. Glottolog uses language identifiers called glottocodes, and we used these glottocodes to determine the language family for each language in our statistical analyses. Some of our data sets already included glottocodes, others included ISO language codes, which were mapped to glottocodes using the `lingtypology` package, and all remaining data sets were manually annotated with glottocodes.

**Deictic day names.** Tent's database[21] includes 157 day-naming systems. A glottocode could not be determined for one language (Natleba), leaving 156 for the regression analysis. Tent reports that some of these systems show *dual symmetry* and use the same forms for days in the past and future. For example, Komba (Papua New Guinea) uses *mara-nan* for '3 days before today' and '3 days after today', and these meanings are distinguished using tense marking. Tent[21] gives Komba, Hindi, and Capanahua (Peru) as examples of systems with dual symmetry, but his data set does not specify the full set of systems with this property. As a result, our analyses do not allow for dual symmetry, and we treat Komba as a 7-term system (i.e., a system that picks out meanings ranging from '3 days before today' to '3 days after today') instead of a 4-term system. Although our theory assumes that categories are connected, dual symmetry provides one example of how disconnected categories can arise for principled reasons.

**Tense.** Velupillai's tense database includes 318 languages. 77 of those languages have no grammatical tense, and an additional 8 make remoteness distinctions, but the number of these distinctions was not specified, meaning that the sizes of these systems could not be determined. We excluded these 85 languages, leaving 233 for analysis. Velupillai's classification allows for zero marking, and a category without overt expression contributes to the size of a tense system as long as it stands in paradigmatic opposition to categories that are marked.

**Seasons.** The seasonal cycle for a climate variable such as temperature can be captured by a function such as {$f : time \rightarrow temperature$} with the form of a sine wave (Fig. 4a). We suggested earlier that the first transformation of the bisected circular structure in Fig. 1 corresponds

to time reversal, but this transformation also corresponds to inverting the temperature axis.

The database of season systems compiled by Kemp et al.[25] includes 54 systems, and we removed one language (Grand Valley Dani) that is reported to have no terms for seasons. Two glottocodes appear twice each in the data, and in both cases, the two systems for those glottocodes are both even. We included only one representative of each glottocode in the regression analysis, yielding 51 systems in total for that analysis.

**Moon phases.** The monthly lunar cycle can be captured by a function {$f$: time → phase} that traces out a helix over time (Fig. 4b). The phase variable ranges over a circular configuration of states, including new moon (bottom of Fig. 4b), first quarter, full moon, and third quarter. Comparing the two panels in Fig. 4 reveals that the domains of seasons and moon phases are not isomorphic. For seasons, the temperature variable corresponds to an interval, not a circle, and a circular structure only arises if we keep track of whether the temperature is rising or falling. For moon phases, the phase variable is intrinsically circular. Although the domains are not isomorphic, both can be represented by the bisected circular structure in Fig. 1.

We know of no existing database of terms for moon phases and therefore compiled our own set of 52 systems. 32 were drawn from the Human Relations Area Files[27] by searching for paragraphs that were classified with the subjects "ordering of time" or "ethnometeorology", and that included either "phase" or "phases" and "lunar" or "moon." The remaining systems were collected by searching Google, Google Scholar and Google Books for combinations of terms and phrases such as "terms for", "phases of" and "moon." Some cultures name every night in a lunar cycle: for example, Kelly and Milone[57] give sequences of 30 Tahitian names and 32 Maori names for nights of the moon. We did not include these cases, and the largest system in our database has 14 terms.

One culture was removed from the analysis because a glottocode could not be reliably identified. An additional two cultures appear twice in our data because different authors give different numbers of terms for these cultures. All of these counts were even, and we included each culture only once in the regression analysis, giving 49 systems overall for that analysis.

**Cardinal directions.** For cardinal systems, we previously noted that the two transformations in Fig. 1 correspond to a reflection in the east-west axis and a rotation of 180°. The rotation can also be replaced by a reflection in the north-south axis, because both alternatives lead to the same transformation group.

Three glottocodes appeared twice in our set of 90 cardinal systems, and we kept one representative of each for the regression analysis. The two systems listed for one of these glottocodes (`waja1257`) have different parities, and we kept the odd-numbered system to be conservative with respect to our hypothesis. Roughly 80% of the 87 languages included in the regression analysis belonged to the same language family (Pama-Nyungyan), and only two other families were represented more than twice in the data set. We therefore ran a standard logistic regression rather than a phylogenetic regression for cardinal directional systems.

**Kinship.** We defined the six subdomains of kinship using the core kin types in Table S2 of Passmore et al.[33]. For siblings, we included 12 kin types: `mB` and `mZ` (which specify the brother and sister of a male speaker), `meB` and `meZ` (which specify the elder brother and sister of a male speaker), `myB` and `myZ` (which specify the younger brother and sister of a male speaker), and the six analogous kin types for a female speaker. We deliberately dropped `mG` and `fG` (sibling of a male and sibling of a female) because we are interested in proper subsets of each subdomain. Lists of the kin types included for the remaining five subdomains are provided in Table S3.

Given a list of kin types, we identified all terms in a language that refer to at least one of these kin types. Kinbank includes many dialectal and orthographic variants, and we wanted to group all terms with the same extensions. We therefore counted all terms that could be used to refer to the same kin types as a single category. For example, three different terms that can each be used to refer to both `meB` and `feB` would be counted as a single category corresponding to "elder brother."

Some languages appear several times in Kinbank with entries attributed to different sources. We kept all of these systems and included a random effect for language in the phylogenetic regression analysis. The number of systems for each subdomain was 1172 (siblings), 1106 (parents' siblings), 869 (grandparents), 449 (grandchildren), 795 (nieces and nephews) and 880 (cousins). The number of languages represented by these systems was 1081 (siblings), 1018 (parents' siblings), 808 (grandparents), 419 (grandchildren), 724 (nieces and nephews) and 803 (cousins).

As a robustness check, we computed distributions over system sizes for kinship systems derived from a second data set compiled by Murdock[58]. The results are reported in Fig. S1, and are similar to the kinship results shown in Fig. 2.

**Australian Aboriginal social classification.** Data for Australian social classification were derived from the AustKin website[38], and our work is consistent with the conditions of use at http://www.austkin.net/index.php?loc=disclaimer. AustKin includes information about ten types of social systems, and we included all of these types except for *phratries* and *totemic groups*. The eight types included were *sections*, *subsections*, *underspecified sections*, *matri-moieties*, *patri-moieties*, *generational moieties*, *matri-semi-moieties*, and *patri-semi-moieties*. Multiple systems are specified for some languages: for example, the Alyawarr data include systems of type *sections* based on four different sources, and an additional system of *subsections*. Figure 2l is based on a set of 331 systems that includes only the largest system for each language. Because all systems were even, we did not run a regression analysis for this domain.

White[59] demonstrates that any society that allows sister exchange or that allows patrilateral cross-cousins to marry must have an even number of categories. Systems with odd-numbered sizes, however, are theoretically possible, and it is reported that the Purum of India have a five-category system[60].

**Non-cardinal directional systems.** The database of Hoffmann et al.[28] distinguishes between multiple types of non-cardinal systems, and

we included those with the following types: *sun only*, *wind only*, *wind and climate*, *climate/weather*, *coast and wind*, *coast only*, *sun coast and river*, *river and sun*, *river only*, *tide*, *spiritual/cultural places of significance*, or *other*. We excluded systems based purely on topographic landmarks (*topographic elevation only*) or vertical elevation (*vertical elevation only*), along with mixed systems that incorporated topographic landmarks or vertical elevation as a component (e.g., *vertical and river* and *topographic elevation and river*). When attempting to determine the sizes of these excluded systems, we often found it difficult to tell whether two terms should be treated as labels for the same concept. For example, the vertical/topographic system of Yadhayekenu includes *akinta* 'up', *anpaantu* 'upwards' and *anpaa/anpima* 'above', and we were unsure whether to treat these terms as labels for one, two or three categories. Because the data set includes multiple non-cardinal systems for some languages, the regression analysis for this domain included a random effect for language. As for our analysis of cardinal systems, however, we used a standard regression rather than a phylogenetic regression because the data are dominated by languages from a single family. The analysis included 141 systems drawn from 82 different languages.

**Color.** We used the version of the World Color Survey released in the `wcs` R package[61]. The database includes labels of color chips provided by individual participants, and for each language, we identified all terms that were provided by at least 60% of participants for at least one chip. The number of such terms was taken as the size of the color-naming system for that language.

**Folk biology.** For each language appearing in Appendix B of Brown[43], we counted the number of entries under the heading "life-form terms." Brown's comments on each language sometimes suggest that one or more of the terms listed is not a true life-form term. For example, he lists three terms for Iwaidja (Australia), but suggests that they are labels for "incipient life-form classes" rather than full-fledged life-form terms. We ignored these distinctions for the purposes of our analysis.

Brown's work has been criticized for relying on a notion of life-form that does not apply across languages, and that leaves out "a large number of highly inclusive categories of intellectual, communicative, ecological, and economic significance" ([62,63], p 344). These concerns threaten some of the conclusions that Brown draws from his data, but seem less relevant to our finding that life-form systems show no systematic tendency towards even or odd sizes.

## Regression models

The phylogenetic regressions used phylogenies derived from Glottolog[55] and were implemented using the `brms` package in R[64]. All priors were set to the default used by the `brms` package, which is a half student-t prior with 3 degrees of freedom, a location of 0 and a scale of 2.5.

For each domain, we started by running four Markov chains for 3000 iterations each, including 1500 burn-in iterations. The target average acceptance probability (`adapt_delta`) was set to 0.95. If `brms` returned any warnings (e.g., warnings about divergent transitions), we increased the length of each chain and/or the acceptance probability until no errors were returned. Our final runs used acceptance probabilities of 0.98 for deictic day names and 0.99 for parents' siblings, grandparents, and grandchildren, and the spatial demonstratives analysis (orientation symmetry) described in the Supplementary Methods. The number of iterations was 6000 for deictic day names and 10,000 for tense, the six kinship subdomains and spatial demonstratives (orientation symmetry). In all remaining cases, the default settings (3000 iterations and acceptance probability of 0.95) produced no warnings.

**Reporting summary**

Further information on research design is available in the Nature Portfolio Reporting Summary linked to this article.

## Data availability

The data used in this study are available at https://github.com/cskemp/parity and archived at https://doi.org/10.5281/zenodo.17429647. Data for day naming, tense, and life forms were transcribed from previous publications by Tent, Velupillai and Brown[21,23,43]. Data for seasons and moon phases are released here for the first time. Data for locational systems were compiled by Hoffmann, Palmer and Gaby[28] and generously provided by these authors. Kinbank data were downloaded from https://github.com/kinbank/kinbank, and the Murdock kinship data analyzed in the Supplementary Methods were downloaded from http://charleskemp.com/kinship/murdockdata.zip. Color data were derived from the `wcsR` package[61]. Data for Australian social classification were derived from the AustKin website[38], and the spatial demonstrative data analyzed in the Supplementary Methods were downloaded from https://github.com/cshnican/spatial_demonstratives.

## Code availability

Code is available at https://github.com/cskemp/parity and archived at https://doi.org/10.5281/zenodo.17429647.

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

## Acknowledgments
I thank Dorothea Hoffmann, Bill Palmer and Alice Gaby for sharing their database of directional systems, and Toby Elmhirst, Alice Gaby, Temuulen Khishigsuren and Terry Regier for comments on the manuscript. This work was supported by ARC FT190100200.

## Author contributions
C.K. designed the study, compiled the data sets, analyzed the data and wrote the paper.

## Competing interests
The author declares no competing interests.
