## [Transparent Peer Review file · Nature Communications]

Symmetry in category systems across languages

Corresponding Author: Professor Charles Kemp

Version 0:

Reviewer comments:

Reviewer #1

(Remarks to the Author)

Review: Even or odd? Symmetry and the structure of category systems across languages

This paper presents a theory of symmetry as it applies to the categorisation of different linguistic domains. The paper proposes that amongst linear 1-dimensional domains, like temperature or life cycles, there is no preference for even or odd categories because there is no focal point to divide a consistent system. Comparably, when a linear structure is bisected with a focal point, there is a tendency to create odd category systems because this symmetry preference asks for the same number of categories before and after the focal point, imposing an odd number of categories. For a similar reason, bisected circular structures, like the circular notion of seasons repeating each year, are more likely to be even. Applied to non-temporal systems, like cardinal directions, also show this symmetry. Finally, a binary feature system will also show a preference for symmetry because each feature imposes an opposition that will result in an even number of relatives. For example, sibling systems are often the combination of binary rules, which result in an even number of categories. In each scenario, this paper argues that the idea of symmetry dictates a preference for odd or even categories, whether that be the division of a particular structure into even categories (resulting in even categories), or the division of sections on either side of a bifurcation (resulting in an odd number of categories).

Overall, I found the breadth of data used to support the symmetry argument convincing. The simplicity and logical nature of a preference for symmetry seems well supported, but I wonder whether the result needs to be more cautiously interpreted, as it is shown throughout the paper, that these results are statistical, and there are always examples to the contrary. In my review, I will tend to focus on the binary features selection, and kinship components of the paper, as that is my speciality, but some other comments interspersed.

To first address bisected circular structures. Although I don't have any complaints about the support for even systems in these structures, the paper seems to overlook why we observe symmetry in these systems. Unlike the bisecting linear system, where it seems logical to have the same number of categories before and after the bisecting point, the bisecting circular structure could feasibly have three bisects and still have symmetry, at least in the mathematical sense, but perhaps there is a different mode of symmetry being invoked here? For example, a circle with three evenly spaced privileged points can be symmetrically divided by drawing a line straight through one point and dividing the other two. Perhaps there is more to be said about the relationship between symmetry and privileged points. The privileged points being of different types (i.e. a cold vs a hot point if we consider seasons), seems like a possible solution the author could explore.

A secondary point on bisected circular structures is that Figure three seems to imply that the symmetry arises from the underlying environmental features (seasons, moon phases). I would like to see the author justify why these support a cognitive argument for the preference of symmetry (which the author seems to be implying is the case across domains on line 28). To me, these particular domains just seem to reflect the symmetry that we observe in the natural setting. In general, it would be nice to have a cognitive theory for why symmetry is observed, which the author seems to be implying is the cause, but to me isn't fully developed.

Turning to binary feature selection, and kinship. Although I think the result is sound, the analysis could be more grounded, I think. As I am sure any specialist would say about any of the domains studied, kinship is complicated!

My first clarification is asking how each category is defined. A typical division of each kinship category could be found in Jones (2010). My summary would be:

- Siblings: 16 total categories; Older vs Younger, Male vs Female, Relative Sex

- Aunts+Uncles: 16 categories; Older / Younger (of connecting relative), Male vs Female, Lineage (mother of father's line), Relative Sex. Note: these terms should be more specifically Parent's siblings – since the English category Aunts + Uncles includes relatives by marriage which I don't think are included here.

- Grandparents: 8 categories; Male vs Female, Distinguish lineage

- Niblings (nieces and nephews): 16 categories: Relative gender, Relative Age, Gender, Lineage

In Figure 1, however, we can see that the maximum number of words for each subset seems to be higher than the maximum number of categories. I think this is likely because the analysis hasn't considered that some languages in Kinbank have multiple sources and therefore have more words than categories (because either sources disagree or offer different orthographies). For example: Latvian is listed as having 10 grandparent terms. If I do this from the raw data, I actually get 12 terms, but as is fairly obvious from the data, most of the words are just variants of kinship terms – like in English where we have Granny, Nanna, Grandmother, Grandma, but we might still say we have one kinship term for Grandmother. Ideally, this wouldn't be the case for Kinbank, but most sources are ambiguous about the terms they record. An approach that might work better with Kinbank data, and would probably result in the same outcome, is using the kinship structure of each language (See <https://osf.io/8wfmU>). You can see in Figure 2A of this paper, cross-referenced with Nerlove and Romney (1967), that the most frequent sibling structures have an even number of categories, and odd-numbered systems are rare. The exception is type 5 which has an elder brother term, an elder sister term, and a younger sibling term, which is the second most common structure in Kinbank, but the most common, third, and fourth systems are all even. The fifth most common is to have a single kinterm for siblings.

A final minor point is that these counts appear to assume that each language is independent of all other languages. It could be the case that languages are from a biased sample of languages that contain even numbered categories, and therefore the result is a sampling problem, rather than an actual feature of language. I think given the scale of the datasets used (maybe aside from the Australian-specific datasets), this problem could be explained away by justifying the likelihood of seeing the predicted preference across such a large sample of languages, oversampling some areas and undersampling others should be balanced out. I would worry that people reading the paper who think that is important will dismiss the result if this point is not addressed at least in brief.

Jones, D. (2010). Human kinship, from conceptual structure to grammar. *Behavioral and Brain Sciences*, 33(5), 367–381. <https://doi.org/10.1017/S0140525X10000890>

(Remarks on code availability)

Reviewer #2

(Remarks to the Author)

review of Even or odd? Symmetry and the structure of category systems across languages

This paper uses a number of test cases from various domains to argue that category systems in language tend to be symmetric given the underlying structure of the domain. This leads to the prediction that some types of category systems should show a preference for a odd number of categories and others an even number of categories. There is some support for the hypothesis provided, though not all test cases provide clear evidence, and in some cases there appears to be a preference for just one type of system. In general, I found the paper intriguing, but not necessarily super convincing. I also have some general questions about how symmetry relates to other cognitive preferences that shape language and about how privilege points can be determined.

(1) A good number of the plots are about kinship, and this is the only binary feature system tested. It wasn't clear to me whether there was a case to be made for separating out each dimension of kinship in this way. And why not look at other binary feature systems? There are typological databases available for the others mentioned. Relatedly, I don't really see what the Aboriginal social classification data get us. There is only a single system (type?) and these languages are all areally and genetically related, no?

(2) There were several points in the manuscript where I wondered whether it was really so simple to determine what the structure of the underlying system should be, i.e., linear vs. bisected-linear. Take seasons for example, where the prediction is odd, but some languages have a bipartition. But this is described in the SI as having a different underlying structure. The same issue came up for me in the description of moon phases. I suppose my question is, how do we separate statistical noise (i.e., the bias is soft, and not all language will conform to the symmetry preference), from differences in category structure, from differences in privileged points? Could it be that some languages differ in whether there IS a privileged point at all in a given domain? It feels to me that more evidence is needed to show that there are universal(ish) category structures

that in fact underlie these domains. That kind of work has been done in the colour domain, and a few other places (some initial work on person systems in <https://escholarship.org/uc/item/69d6v383>), but without that, I'm not so sure it makes sense to assign a given domain a given structure in the way that this work does.

(3) Perhaps related to the above comment, I also wondered about the relationship between defaults (or unmarked forms) and privileged points. For example in the non-cardinal systems section, there is brief mention of a language that has a term for "other side of the river" but not "same side of the river". Similar question came up in the tense section, where perhaps present tense is the default. What role if any do defaults play here?

(4) A big question I had while reading this was how symmetry relates to other concepts like monotonicity, connectedness, structural iconicity, and isomorphism. These have all been argued to shape semantic systems, and they all feel related to, but not necessarily the same as symmetry. Perhaps structural iconicity and isomorphism are most closely related to symmetry? I also wondered whether some instances of symmetry are explain by compositionality, i.e., when languages create a new form for use with one end of a scale, it is then available for use at the other end, So why not? If you see what I mean. I would appreciate more discussion of this, and indeed I think this would make the paper more interesting, though maybe more for specialists (see my next comment).

(5) My final comment is about the abstract/introduction and the audience. First, for me the abstract was not a great summary of the paper. I would not understand what the paper is claiming, or showing based on what it says. What is "the parity of category systems"? What is the relation between symmetry and odd and even? What is "the principle of symmetry" and why do the results "therefore provide evidence"? I don't see how the two halves of this sentence connect. I also found the introduction to the paper tough going, particularly section 1. The very beginning gives not much sense for what is at stake, or how symmetry is related to simplicity (or other concepts, see above). The discussion around Figure 1 is really not very clear, only the linear structure is really discussed in the text. There are a few critical sentences that need more scaffolding, "All of these categories are connected...(line 63)", "...if it is left unchanged...(line 68)". I'm wondering whether it makes sense to use some of the examples in the text that follows to help the reader here. The rest of the paper is more clearly written. My general feeling is that paper is in some sense presenting some initial findings, that would benefit from more nuanced discussion, and therefore might be better in a more specialised journal.

(Remarks on code availability)

Reviewer #3

(Remarks to the Author)

The paper is highly interesting and based on a very clear idea, that human systems are symmetrical, something that can be explained by their functionality and salience. Systems are symmetrical in different ways, either even or odd, and that there is a tendency for systems globally to be either of the one or the other type. In the paper, these questions are investigated by mass data and Bayesian methodologies. I think the topic is highly interesting and presented in an appropriate way, even if I believe that there are several shortcomings in the study, that should be accounted for before publication.

1. The sources: the data stems from various sources, of which some are global, some more restricted. This is mentioned in the study. However, of the global sources, it is not clear how the distributions of data is in relation to areas and language families (at least not in the material as it is presented). I guess that it would be possible to extract this information from the raw data sets, but the supplementary material should have all available statistics on the data sets used for the study. As it is now, data sets stem from a number of different sources, and it is fully unclear in which way the results are representative if the data sets are not compared to each other. Therefore, the data needs a significant review and presentation by the author. If the information is there somewhere, the paper needs to point out where the reader can find this information.

2. The model: Results of global statistics involving language systems must account for possible autocorrelation or other skewing of results related to an area- or family-bias. This can be accounted for in different ways by the model, most appropriately by a model that builds in family or area. A more advanced model would build in a phylogeny, but I assume that this is not possible in the cases presented here. In any case, it is uncertain whether - and to what extent - the data represents similar systems because of this underlying factor (area- or family-bias, as described in previous paragraph).

Further, I think that the graphs are very uninformative. All different systems are presented in one graph (Figure 2) and it is entirely unclear what represents what etc. This can only be concluded by reading the text carefully, which is no advantage. Figure captions should carefully describe what they represent and how they reflect the results of the study.

(Remarks on code availability)

Version 1:

Reviewer comments:

Reviewer #1

(Remarks to the Author)

Thank-you for addressing and clarifying my comments. I think this is a highly interesting paper, and I commend the unification of domains under a single, simple idea, as well as the supporting quantitative analysis, which utilises the breadth of datasets that are increasingly available. I look forward to seeing the reception of the paper more broadly once it is

published.

(Remarks on code availability)

I didn't review the code the first time, but I have looked at it this time and found it, at least qualitatively, well described and documented. I had various problems with running the BRMS models, but that is due to my own technological limitations (and the limitations imposed on my by Apple!). I've listed some minor notes here that might help others reproduce the analysis, but these shouldn't delay production.

- The package wcs is a custom package. It is worth putting a comment showing how this package can be installed (`remotes::install_github("jvosten/wcs")`) - particularly since all other packages are on CRAN.
- There are some utility functions found throughout the code. It might make sense to move these into the regression functions chunk and call it "functions".
- Given it is unlikely that this code will work forever, it might also benefit from some more descriptive comments throughout.

And a final comment unrelated to the paper, but that might be helpful in the future: There is a note that the regression models take some time to run. I assume this is the brms MCMC process. A non-MCMC Bayesian alternative is INLA, which has a lot of speed improvements (although 2 hours isn't all that long either I suppose!).

Reviewer #2

(Remarks to the Author)

I thank the author for engaging thoroughly with my comments. I'm satisfied with the revised manuscript.

(Remarks on code availability)

Reviewer #3

(Remarks to the Author)

My comments/concerns have been met in this new version of the paper.

(Remarks on code availability)

RESPONSE TO REVIEWERS

I thank all reviewers for their careful consideration of my manuscript. I have attempted to address all points raised and believe that the manuscript is stronger as a result.

Reviewer #1 (Remarks to the Author):

Review: Even or odd? Symmetry and the structure of category systems across languages

This paper presents a theory of symmetry as it applies to the categorisation of different linguistic domains. The paper proposes that amongst linear 1-dimensional domains, like temperature or life cycles, there is no preference for even or odd categories because there is no focal point to divide a consistent system. Comparably, when a linear structure is bisected with a focal point, there is a tendency to create odd category systems because this symmetry preference asks for the same number of categories before and after the focal point, imposing an odd number of categories. For a similar reason, bisected circular structures, like the circular notion of seasons repeating each year, are more likely to be even. Applied to non-temporal systems, like cardinal directions, also show this symmetry. Finally, a binary feature system will also show a preference for symmetry because each feature imposes an opposition that will result in an even number of relatives. For example, sibling systems are often the combination of binary rules, which result in an even number of categories. In each scenario, this paper argues that the idea of symmetry dictates a preference for odd or even categories, whether that be the division of a particular structure into even categories (resulting in even categories), or the division of sections on either side of a bifurcation (resulting in an odd number of categories).

Overall, I found the breadth of data used to support the symmetry argument convincing. The simplicity and logical nature of a preference for symmetry seems well supported, but I wonder whether the result needs to be more cautiously interpreted, as it is shown throughout the paper, that these results are statistical, and there are always examples to the contrary. In my review, I will tend to focus on the binary features selection, and kinship components of the paper, as that is my speciality, but some other comments interspersed.

Thanks for these positive comments. I have attempted to acknowledge the statistical nature of the predictions throughout – for example the abstract uses the phrase “tend to” three times, and the opening paragraph of the discussion explicitly acknowledges that the predictions are “statistical tendencies”

To first address bisected circular structures. Although I don't have any complaints about the support for even systems in these structures, the paper seems to overlook why we observe

symmetry in these systems. Unlike the bisecting linear system, where it seems logical to have the same number of categories before and after the bisecting point, the bisecting circular structure could feasibly have three bisects and still have symmetry, at least in the mathematical sense, but perhaps there is a different mode of symmetry being invoked here? For example, a circle with three evenly spaced privileged points can be symmetrically divided by drawing a line straight through one point and dividing the other two. Perhaps there is more to be said about the relationship between symmetry and privileged points. The privileged points being of different types (i.e. a cold vs a hot point if we consider seasons), seems like a possible solution the author could explore.

Thanks for suggesting that more discussion of privileged points is needed. The supporting material contains an extended discussion of privileged points and the role that they play in the theory. I assume that privileged points form part of the space (second row of Figure 1) over which categories are defined – in other words, the privileged points form part of people’s mental representation of a given domain. From this perspective, a circular structure with three privileged points is possible in principle, but I propose that people do not represent the relevant domains (seasons, lunar phases, cardinal directions) in this way.

As suggested in the supplementary material, for seasons the privileged points could correspond to the hottest and coldest time of the year, or the two equinoxes. Either possibility produces the prediction that season systems tend to have an even number of categories.

A secondary point on bisected circular structures is that Figure three seems to imply that the symmetry arises from the underlying environmental features (seasons, moon phases). I would like to see the author justify why these support a cognitive argument for the preference of symmetry (which the author seems to be implying is the case across domains on line 28). To me, these particular domains just seem to reflect the symmetry that we observe in the natural setting. In general, it would be nice to have a cognitive theory for why symmetry is observed, which the author seems to be implying is the cause, but to me isn’t fully developed.

Thanks for raising this point. I have added a paragraph to the discussion (line 297) that considers whether the symmetries arise from the world or the mind. This paragraph acknowledges that the results do not resolve this question, and suggests that lab studies could be used to see whether people create category systems with greater degrees of symmetry than are present in the environment.

Turning to binary feature selection, and kinship. Although I think the result is sound, the analysis could be more grounded, I think. As I am sure any specialist would say about any of the domains studied, kinship is complicated!

My first clarification is asking how each category is defined. A typical division of each kinship category could be found in Jones (2010). My summary would be:

- Siblings: 16 total categories; Older vs Younger, Male vs Female, Relative Sex

- Aunts+Uncles: 16 categories; Older / Younger (of connecting relative), Male vs Female, Lineage (mother of father's line), Relative Sex. Note: these terms should be more specifically Parent's siblings – since the English category Aunts + Uncles includes relatives by marriage which I don't think are included here.

- Grandparents: 8 categories; Male vs Female, Distinguish lineage

- Niblings (nieces and nephews): 16 categories: Relative gender, Relative Age, Gender, Lineage

In Figure 1, however, we can see that the maximum number of words for each subset seems to be higher than the maximum number of categories. I think this is likely because the analysis hasn't considered that some languages in Kinbank have multiple sources and therefore have more words than categories (because either sources disagree or offer different orthographies). For example: Latvian is listed as having 10 grandparent terms. If I do this from the raw data, I actually get 12 terms, but as is fairly obvious from the data, most of the words are just variants of kinship terms – like in English where we have Granny, Nanna, Grandmother, Grandma, but we might still say we have one kinship term for Grandmother. Ideally, this wouldn't be the case for Kinbank, but most sources are ambiguous about the terms they record. An approach that might work better with Kinbank data, and would probably result in the same outcome, is using the kinship structure of each language (See <https://osf.io/8wfmU>). You can see in Figure 2A of this paper, cross-referenced with Nerlove and Romney (1967), that the most frequent sibling structures have an even number of categories, and odd-numbered systems are rare. The exception is type 5 which has an elder brother term, an elder sister term, and a younger sibling term, which is the second most common structure in Kinbank, but the most common, third, and fourth systems are all even. The fifth most common is to have a single kinterm for siblings.

Thanks for the general suggestion here and for the pointer to Passmore (2023). The Methods (line 443) and Supplementary Material (p 1) now describe how the subdomains of kinship are defined, and I have followed your suggestion and changed “aunts+uncles” to “parents' siblings”.

I now acknowledge around line 451 that some categories can have multiple synonymous labels, and my approach aims to count categories rather than labels. I have changed my code, however, to better achieve this goal – for example, the code is no longer sensitive to capitalization, and I identified and fixed a couple of other bugs. Currently the 12 Latvian terms for grandparents are organized into seven categories:

- 1) vectētiņš, vectēvs, vectētiņš : mFF, mMF, fFF, fMF
- 2) vecmāmiņa, vecmāmuļa : mFM, mMM, fFM, fMM
- 3) tēva tēvs: mFF
- 4) vecmate, vecāmāte: mFM, mMM
- 5) vacaistevs, vecaistēvs: mFF, mMF
- 6) mātes māte: mMM
- 7) Vecmāmiņa: mFM

Ideally the count here would be six: vecmāmiņa and Vecmāmiņa are distinguished here because the two have different diacritics on the n. But I think that a general policy of ignoring diacritics would be a mistake because they may carry important distinctions for some languages. The real problem here is an issue of missing data: Vecmāmiņa would have been merged with category 2 if Vecmāmiņa had been paired with MM, fFM, fMM, but in Kinbank it is only paired with mFM.

To ensure that missing data has not distorted the results, I added a second kinship analysis using a different data set compiled by Murdock. The results (Figure S1) are compatible with the kinship results, suggesting that the conclusions about kinship are robust. I did not run phylogenetic regressions based on the Murdock data because this data set does not include glottocodes.

Instead of using the Murdock data set, I considered implementing the “structural vector” approach described by Passmore (2023) in <https://osf.io/8wfmj> . I found the code released along with this paper helpful, and incorporated some of it into my own scripts. But I ended up using the Murdock data for two reasons. First, analysing a second data set seemed a useful way to demonstrate robustness. Second, if I understand correctly, the “structural vector” approach takes the first term listed in Kinbank for each kin type, and this does not seem ideal. As far as I can tell there is no guarantee that the first term listed in Kinbank is actually the “best” or “default” term for that kin type.

A final minor point is that these counts appear to assume that each language is independent of all other languages. It could be the case that languages are from a biased sample of languages that contain even numbered categories, and therefore the result is a sampling problem, rather than an actual feature of language. I think given the scale of the datasets used (maybe aside from the Australian-specific datasets), this problem could be explained away by justifying the likelihood of seeing the predicted preference across such a large sample of languages, oversampling some areas and undersampling others should be balanced out. I would worry that people reading the paper who think that is important will dismiss the result if this point is not addressed at least in brief.

Thanks for raising this point. I attempt to address it by reporting estimates derived from phylogenetic regression models that control for language relatedness (Figure 3). I have also adjusted the text (line 115) to explain that the regression models are included for this purpose.

Jones, D. (2010). Human kinship, from conceptual structure to grammar. Behavioral and Brain Sciences, 33(5), 367–381. <https://doi.org/10.1017/S0140525X10000890>

I now cite this paper on line 202.

Reviewer #2 (Remarks to the Author):

review of Even or odd? Symmetry and the structure of category systems across languages

This paper uses a number of test cases from various domains to argue that category systems in language tend to be symmetric given the underlying structure of the domain. This leads to the prediction that some types of category systems should show a preference for a odd number of categories and others an even number of categories. There is some support for the hypothesis provided, though not all test cases provide clear evidence, and in some cases there appears to be a preference for just one type of system. In general, I found the paper intriguing, but not necessarily super convincing. I also have some general questions about how symmetry relates to other cognitive preferences that shape language and about how privilege points can be determined.

(1) A good number of the plots are about kinship, and this is the only binary feature system tested. It wasn't clear to me whether there was a case to be made for separating out each dimension of kinship in this way.

Considering subsystems (e.g. siblings, grandparents, grandchildren, etc) separately is relatively common in work on kinship – for example, the supplementary material now includes results based on a dataset compiled by Murdock (1970) that is organized in terms of subsystems. Following this practice here has the advantage of allowing my results to be compared with previous work that focuses on subsystems (as R1 did above when comparing with previous work by Passmore and Nerlove and Romney on siblings).

And why not look at other binary feature systems? There are typological databased available for the others mentioned.

Thanks for this suggestion. I have added an analysis of spatial demonstratives to the supplementary material. I considered the other domains listed but all of them present various complications. For example, the representation of personal pronoun systems used by Zaslavsky

et al (2021) includes 11 elements and cannot be characterized as a simple product of a person feature (speaker, addressee, and other) and a number feature (one, two, more than two), because this product would generate only 9 elements. Similarly, Haspelmath's (1997) feature-based representation of indefinite pronoun systems includes one feature (scale reversal) that depends on another (scalar endpoint), which makes these representations complicated to work with. Given the relatively large number of domains already considered I feel that analyzing two feature-based domains (ie kinship and spatial demonstratives) is sufficient.

Relatedly, I don't really see what the Aboriginal social classification data get us. There is only a single system (type?) and these languages are all areally and genetically related, no?

These languages are all related, but there is more than one system – Figure 2I shows that the data set includes systems of several different sizes (ie some make finer-grained distinctions than others). In my mind the social classification data are valuable for two main reasons. First, the absence of odd-numbered systems in this domain provides a counterpoint to other domains for which symmetry is a statistical tendency. Second, the group-theoretic analysis of social classification included in Levi-Strauss (1949) is probably the most influential prior attempt to analyze category systems using symmetry principles. Including social classification as a domain provides a way to connect with that work, which illustrates that symmetry has implications that go beyond parity alone.

(2) There were several points in the manuscript where I wondered whether it was really so simple to determine what the structure of the underlying system should be, i.e., linear vs. bisected-linear. Take seasons for example, where the prediction is odd, but some languages have a bipartition. But this is described in the SI as having a different underlying structure. The same issue came up for me in the description of moon phases. I suppose my question is, how do we separate statistical noise (i.e., the bias is soft, and not all language will conform to the symmetry preference), from differences in category structure, from differences in privileged points? Could it be that some languages differ in whether there IS a privileged point at all in a given domain? It feels to me that more evidence is needed to show that there are universal(ish) category structures that in fact underlie these domains. That kind of work has been done in the colour domain, and a few other places (some initial work on person systems in <https://escholarship.org/uc/item/69d6v383>), but without that, I'm not so sure it makes sense to assign a given domain a given structure in the way that this work does.

I agree that the structures proposed in the current paper should be viewed as working hypotheses rather than well-established universals, and have added a paragraph to the Discussion that makes this point (line 288). I see the current paper as analogous to some of the early work on efficient communication and colour. Initial studies assumed that need probabilities (ie the frequency with which people talk about different hues) were constant across cultures, but subsequent work has relaxed that assumption.

(3) Perhaps related to the above comment, I also wondered about the relationship between defaults (or unmarked forms) and privileged points. For example in the non-cardinal systems section, there is brief mention of a language that has a term for "other side of the river" but not "same side of the river". Similar question came up in the tense section, where perhaps present tense is the default. What role if any do defaults play here?

Thanks for raising this. Of the domains considered, I agree that tense and non-cardinal directional systems are the two that raise questions about zero marking. I now say that Velupillai's tense data incorporates categories without overt expression (line 399), and that defaults may explain why only one member of a spatial opposition is named (line 247).

(4) A big question I had while reading this was how symmetry relates to other concepts like monotonicity, connectedness, structural iconicity, and isomorphism. These have all been argued to shape semantic systems, and they all feel related to, but not necessarily the same as symmetry. Perhaps structural iconicity and isomorphism are most closely related to symmetry? I also wondered whether some instances of symmetry are explain by compositionality, i.e., when languages create a new form for use with one end of a scale, it is then available for use at the other end, So why not? If you see what I mean. I would appreciate more discussion of this, and indeed I think this would make the paper more interesting, though maybe more for specialists (see my next comment).

Thanks for this question. I have added two new paragraphs to the discussion that consider symmetry of linguistic form and mention both iconicity and compositionality (line 307). Compositionality is also now mentioned in the following paragraph (line 330).

I have not discussed monotonicity and connectedness because do not see clear links between symmetry and these principles. In the case of monotonicity, I'm not sure how to apply this principle to structures other than the linear or bisected linear structures, and for these structures I think that monotonicity may not be related to symmetry. Monotonicity predicts, I think, that the structure will be divided into two pieces by making a single cut somewhere along the dimension – that way one of the resulting categories will be upward monotonic and the other will be downward monotonic. But monotonicity does not require that the single cut occurs in the middle of the dimension, which means that monotonicity does not imply symmetry.

For connectedness, I think that symmetry does not imply connectedness (consider a symmetric system that uses one term for both ends of a dimension and a second term for the middle), and that connectedness does not imply symmetry. So I'm not sure what to say about connectedness other than the fact that for connected domains, my predictions about parity rely on the assumption that categories are connected (line 65).

(5) My final comment is about the abstract/introduction and the audience. First, for me the abstract was not a great summary of the paper. I would not understand what the paper is

claiming, or showing based on what it says. What is "the parity of category systems"? What is the relation between symmetry and odd and even? What is "the principle of symmetry" and why do the results "therefore provide evidence"? I don't see how the two halves of this sentence connect. I also found the introduction to the paper tough going, particularly section 1. The very beginning gives not much sense for what is at stake, or how symmetry is related to simplicity (or other concepts, see above). The discussion around Figure 1 is really not very clear, only the linear structure is really discussed in the text. There are a few critical sentences that need more scaffolding, "All of these categories are connected...(line 63)", "...if it is left unchanged...(line 68)". I'm wondering whether it makes sense to use some of the examples in the text that follows to help the reader here. The rest of the paper is more clearly written. My general feeling is that paper is in some sense presenting some initial findings, that would benefit from more nuanced discussion, and therefore might be better in a more specialised journal.

Thanks for pointing out that the abstract and introduction could be improved. I have attempted to address as follows:

- (i) I have rewritten both the abstract and the introduction in an attempt to be clearer – for example, the term “parity” is no longer mentioned in the abstract, and I try to highlight the key idea that symmetry shapes categorization across cultures.
- (ii) When Figure 1 is first introduced, I now say explicitly say that the initial discussion will focus on the linear structure, and that the other structures will be introduced in subsequent sections (line 56).
- (iii) I have adjusted the critical sentences (formerly lines 63 and 68) and added concrete examples that illustrate what it means for a category to be connected and for a category system to be symmetric (see lines 67 and 73)

Reviewer #3 (Remarks to the Author):

The paper is highly interesting and based on a very clear idea, that human systems are symmetrical, something that can be explained by their functionality and salience. Systems are symmetrical in different ways, either even or odd, and that there is a tendency for systems globally to be either of the one or the other type. In the paper, these questions are investigated by mass data and Bayesian methodologies. I think the topic is highly interesting and presented in an appropriate way, even if I believe that there are several shortcomings in the study, that should be accounted for before publication.

Thanks for these positive comments!

1. The sources: the data stems from various sources, of which some are global, some more restricted. This is mentioned in the study. However, of the global sources, it is not clear how

the distributions of data is in relation to areas and language families (at least not in the material as it is presented). I guess that it would be possible to extract this information from the raw data sets, but the supplementary material should have all available statistics on the data sets used for the study. As it is now, data sets stem from a number of different sources, and it is fully unclear in which way the results are representative if the data sets are not compared to each other. Therefore, the data needs a significant review and presentation by the author. If the information is there somewhere, the paper needs to point out where the reader can find this information.

Thanks for raising this. The supplementary material now opens with tables documenting the distribution of languages across areas and families, and the main text refers to these tables on line 108.

2. The model: Results of global statistics involving language systems must account for possible autocorrelation or other skewing of results related to an area- or family-bias. This can be accounted for in different ways by the model, most appropriately by a model that builds in family or area. A more advanced model would build in a phylogeny, but I assume that this is not possible in the cases presented here. In any case, it is uncertain whether - and to what extent - the data represents similar systems because of this underlying factor (area- or family-bias, as described in previous paragraph).

To account for genetic relatedness, the previous version used mixed effects regression models with a random intercept for language family. Following your suggestion, I have now replaced these with phylogenetic regression models.

Further, I think that the graphs are very uninformative. All different systems are presented in one graph (Figure 2) and it is entirely unclear what represents what etc. This can only be concluded by reading the text carefully, which is no advantage. Figure captions should carefully describe what they represent and how they reflect the results of the study.

I have now broken Figure 2 into two separate figures (Figs 2 and 3), and the current Figure 2 places the size distribution and parity distribution for a domain side-by-side in a single panel instead of separating them. I have also adjusted and expanded the captions, and I hope that these changes have made the figures more informative.

RESPONSE TO REVIEWERS

I thank all reviewers for engaging with the revision of my manuscript. All reviewers seemed satisfied with the paper itself, and Reviewer 1 made some suggestions about the code.

I didn't review the code the first time, but I have looked at it this time and found it, at least qualitatively, well described and documented. I had various problems with running the BRMS models, but that is due to my own technological limitations (and the limitations imposed on my by Apple!). I've listed some minor notes here that might help others reproduce the analysis, but these shouldn't delay production.

- The package wcs is a custom package. It is worth putting a comment showing how this package can be installed (remotes::install_github("jvosten/wcs")) - particularly since all other packages are on CRAN.

The code uses the R package renv which should allow all packages to be easily installed using renv::restore(). I've now documented this in the top level README. That README also now specifically mentions the wcs package and explains how it can be installed without renv if needed.

- There are some utility functions found throughout the code. It might make sense to move these into the regression functions chunk and call it "functions".

Thanks for the suggestion. I've added a new chunk for utility functions after the regression functions chunk.

- Given it is unlikely that this code will work forever, it might also benefit from some more descriptive comments throughout.

I have added some comments in an attempt to make things clearer.

And a final comment unrelated to the paper, but that might be helpful in the future: There is a note that the regression models take some time to run. I assume this is the brms MCMC process. A non-MCMC Bayesian alternative is INLA, which has a lot of speed improvements (although 2 hours isn't all that long either I suppose!).

Thank you for the pointer to INLA – I will keep it in mind for future projects!

In addition to the changes just mentioned I also updated all R packages to their current versions and made some minor changes to avoid functions that are now deprecated. As mentioned in the manuscript, all code is now publicly available on GitHub (

<https://github.com/cskemp/parity>) and archived under a DOI on Zenodo (<https://doi.org/10.5281/zenodo.17429647>).

Thank you for the opportunity to improve the code,

Charles Kemp